# Gosha-Jinki-Gan Reduces Inflammation in Chronic Ischemic Stroke Mouse Models by Suppressing the Infiltration of Macrophages

**DOI:** 10.3390/biom15081136

**Published:** 2025-08-06

**Authors:** Mingli Xu, Kaori Suyama, Kenta Nagahori, Daisuke Kiyoshima, Satomi Miyakawa, Hiroshi Deguchi, Yasuhiro Katahira, Izuru Mizoguchi, Hayato Terayama, Shogo Hayashi, Takayuki Yoshimoto, Ning Qu

**Affiliations:** 1Department of Immunoregulation, Institute of Medical Science, Tokyo Medical University, Tokyo 160-8402, Japan; yoshitomi@med.showa-u.ac.jp (M.X.); rosesm@tokyo-med.ac.jp (S.M.); deguchi.toui@gmail.com (H.D.); yasuhiro@tokyo-med.ac.jp (Y.K.); imizo@tokyo-med.ac.jp (I.M.); yoshimot@tokyo-med.ac.jp (T.Y.); 2Department of Anatomy, Tokai University School of Medicine, Isehara 259-1193, Japan; suyama@tokai.ac.jp (K.S.); nagahori.kenta.t@tokai.ac.jp (K.N.); kiyoshima@tokai.ac.jp (D.K.); terahaya@tokai.ac.jp (H.T.); sho5-884@umin.ac.jp (S.H.)

**Keywords:** chronic ischemic stroke, middle cerebral artery occlusion (MCAO), Gosha-jinki-gan (TJ107), inflammation, macrophage, glial fibrillary acidic protein (GFAP)

## Abstract

Ischemic stroke is a primary cause of cerebrovascular diseases and continues to be one of the leading causes of death and disability among patients worldwide. Pathological processes caused by vascular damage due to stroke occur in a time-dependent manner and are classified into three categories: acute, subacute, and chronic. Current treatments for ischemic stroke are limited to effectiveness in the early stages. In this study, we investigated the therapeutic effect of an oriental medicine, Gosha-jinki-gan (TJ107), on improving chronic ischemic stroke using the mouse model with middle cerebral artery occlusion (MCAO). The changes in the intracerebral inflammatory response (macrophages (*F4/80*), *TLR2**•**4*, *IL-23*, *IL-17*, *TNF-α*, and *IL-1β*) were examined using real-time RT-PCR. The MCAO mice showed the increased expression of glial fibrillary acidic protein (GFAP) and of *F4/80*, *TLR2*, *TLR4*, *IL-1β*, *TNF-α*, and *IL-17* in the brain tissue from the MCAO region. This suggests that they contribute to the expansion of the ischemic stroke infarct area and to the worsening of the neurological symptoms of the MCAO mice in the chronic phase. On the other hand, the administration of TJ107 was proven to reduce the infarct area, with decreased GFAP expression, suppressed macrophage infiltration in the brain, and reduced *TNF-α*, *IL-1β*, and *IL-17* production compared with the MCAO mice. This study first demonstrated Gosha-jinki-gan’s therapeutic effects on the chronic ischemic stroke.

## 1. Introduction

Ischemic stroke is a primary cause of cerebrovascular diseases and represents one of the major causes of death and disability in patients around the world [1,2]. The pathological processes caused by vascular damage due to strokes occur in a time-dependent manner and are categorized as acute (from a few minutes to a few hours), subacute (from a few hours to a few days), and chronic (from a few days to a few months) [3]. The metabolic disturbances and excitotoxicity contribute to the progression of neuronal damage in the acute phase; inflammation and cell death are the dominant events in the subacute period; and brain repair is the major event and includes microgliosis, glial scar formation, and angiogenesis in the chronic phase, as well as neuronal regeneration [3,4,5].

The main mechanisms of damage in cerebral infarction are different in the acute, subacute, and chronic period, and currently, activation of the immune system is recognized as an important factor in all stages of stroke pathophysiology, especially in the long-term regenerative processes [6,7]. The release of danger-associated molecular patterns (DAMPs), among the initial events after arterial occlusion, are the local expression of proinflammatory cytokines, subsequent expression of endothelial adhesion molecules, and breakdown of the blood–brain barrier (BBB) [7]. A study of a cerebral ischemia model revealed an inflammatory process centered on macrophages during the acute phase several days after the onset of cerebral infarction. In response to stimulation by DAMPs derived from dead cells, brain-infiltrating macrophages are activated in a toll-like receptor (TLR) 2/4-dependent manner, producing inflammatory cytokines such as IL-23. IL-23 induces IL-17 production from T cells that infiltrate later, promoting inflammation in the subacute phase, which contributes to the expansion of the infarct and the worsening of neurological symptoms [8].

Current treatments for ischemic stroke are limited in effectiveness to the early stages of onset. When autologous cultured bone marrow mesenchymal stem cells (MSCs) were administered intravenously to patients with cerebral infarction, significant recovery of motor function was observed [9]. The preconditioned MSCs administration promoted tissue regeneration, reduced neuroinflammation, and increased angiogenesis and myelinization in some rodent models of stroke, traumatic brain injury, and spinal cord injury. The mechanism of this involves neurotrophic and protective action by cytokines, angiogenesis (restoration of cerebral blood flow), and nerve regeneration, and the recovery of targeted functions can be diverse, including not only motor functions but also higher brain functions and excretion [10,11]. However, there is currently no treatment available to improve the established neurological sequelae occurring in the chronic phase. In a group of ischemic stroke survivors aged 65 and older, observed over the 6 months following the stroke, 50% had hemiparesis, 46% had cognitive deficits, and approximately 30% were dependent on others for activities of daily living [12]. There is a need to develop new therapeutic agents that are effective in the chronic phase after disease onset. Recently, there has been considerable attention focused on traditional medicine, as it represents a source of neuroprotective compounds. Natural products, especially medicinal plants, may offer an ideal option for the development of safe and effective drugs for the treatment of stroke [13].

One such oriental medicine, Gosha-jinki-gan (TJ107), which is composed of 10 herbal ingredients (*Rehmanniae radix*, *Achyranthis radix*, *Corni fructus*, *Dioscoreae rhizome*, *Plantaginis semen*, *Alismatis rhizome*, *Hoelen*, *Moutan cortex*, *Cinnamomi cortex*, and *processed Aconite tuber*), is widely used in Japan and China, especially in elderly patients, to treat symptoms such as neuralgia, lower back pain, numbness, and neuropathies. Recently, we demonstrated that TJ107 can completely restore the damaged seminiferous epithelium by normalizing macrophage migration and reducing the expression of TLR2 and 4. It also promoted recovery from severe azoospermia following busulfan treatment in mice [14,15]. In this study, we will investigate the effect of administering TJ107 on chronic ischemic stroke improvement using the mouse model with middle cerebral artery occlusion (MCAO).

## 2. Materials and Methods

### 2.1. Animals

CB-17/Icr-+/+Jcl (CB-17) and MCAO (Clea Japan, Inc. (Tokyo, Japan), CB-17 mice performed by electrocoagulation, leading to the disconnection of the distal part of left middle cerebral artery (MCA)). Briefly, the mouse was anesthetized with halothane in a mixture of 70% nitrous oxide and 30% oxygen. The temperature of the head was maintained at 36 °C, and using a heating lamp, cerebral blood flow (CBF) was measured before and during ischemia by laser Doppler flowmetry on the same side of the parietal bone (∼1–2 mm posterior to bregma). The resting CBF value of each mouse was regarded as its baseline, and changes in CBF after the induction of cerebral ischemia were expressed as a percentage of the resting value. The cerebral infarction model was created by directly applying electrical coagulation to the left MCA. During left MCA ligature, a decrease of over 60% in CBF was confirmed by laser Doppler flowmetry; male mice aged 6 weeks were purchased from Clea Japan, Inc. (Tokyo, Japan). Animals had free access to food and were maintained in the Animal Laboratory of Support Center for Medical Research and Education, Tokai University (Tokai University Animal Committee; No. 214029-01042021 and No. 225016-01042022) under controlled temperatures of 22 to 24 °C and relative humidity of 50 to 60%, with a 12 h light–dark cycle.

### 2.2. Preparation for Gosha-Jinki-Gan (TJ107) Diet

TJ107 (extract granular material in powder form; No. 2230107010) was produced by Tsumura & Co. (Tokyo, Japan) in accordance with Japanese and international manufacturing guidelines. As a standardized prescription drug, TJ107 contains 10 types of herbal ingredients: *Rehmanniae radix* 5 g; *Achyranthis radix* 3 g, *Corni fructus* 3 g, *Dioscoreae rhizome* 3 g, *Plantaginis semen* 3 g, *Alismatis rhizome* 3 g, *Hoelen* 3 g, *Moutan cortex* 3 g, *Cinnamomi cortex* 1 g, and processed *Aconite tuber* 1 g. The TJ107 diet is based on a standard mouse diet (MF diet: crude protein 23.1% (*w*/*w*), crude fat 5.1%, crude ash 5.8%, crude fiber 2.8%, and nitrogen-free extract and mineral mixture 55.3%) and included 5.4% (*w*/*w*) TJ107 extract by Oriental Yeast Co., Ltd. (Tokyo, Japan) [14,15].

### 2.3. Experimental Design

Normal CB-17 mice were divided into a control group (*n* = 10; mice were given the standard MF diet until 63 days after arrival date, Day 0) and TJ107 group (*n* = 10; mice were given the TJ107 diet until 63 days after arrival date, Day 0). MCAO mice arrived one week after surgery and were divided randomly into the MCAO group (*n* = 10; mice that received surgery and were given the standard MF diet until 63 days after arrival date, Day 0) and the MCAO + TJ107 group (*n* = 10; mice that received surgery and were given the TJ107 diet until 63 days after arrival date, Day 0). General condition, food intake (3 g diet/mouse/day; 0.65 mg TJ107/kg/day), and the body weight of all mice were recorded at 7-day intervals from Day 0 to Day 63 after the injection. All experimental protocols in this research were conducted in accordance with the guidelines of the National Institutes of Health and were approved by the Tokai University Animal Care and Use Committee (approval code 225016 on 17 April 2022).

### 2.4. Histopathological Examination

Sodium pentobarbital (FUJIFILM (Osaka, Japan)) (50 mg/kg) was randomly administered intraperitoneally to the mice (*n* = 5), and perfusion was performed via cardiac puncture with 4% PFA, as described previously [16,17,18]. The whole brain was removed and immobilized with 4% PFA for 24 h. They were then embedded in paraffin and coronal slices sliced into 5 µm sections. Continuous sections from the central part of the forebrain were subjected to hematoxylin and eosin (H&E) staining and immunohistochemistry. For each section, the infarct area was measured and analyzed using Image-j software (ImageJ-win64) (FIJI; Wayne Rasband, NIH, Bethesda, MD, USA). The infarct volume (µm^3^) was calculated as the average of five consecutive sections.

After deparaffinization, sections with a thickness of 5 μm were heated in a microwave for 10 min using citrate buffer (pH 6.0; Abcam, Cambridge, UK) for epitope retrieval. After that, the samples were incubated for 2 h with rat-derived anti-mouse F4/80 monoclonal antibodies (immunohistochemical staining of macrophages; Abcam, 1:100 dilution) or rabbit anti-mouse, anti-glial fibrillary acidic protein (GFAP) monoclonal antibodies (immunohistochemical staining of astrocytes; Abcam, 1:2000 dilution). The sections were incubated for 30 min with rabbit anti-rat or goat anti-rabbit IgG (Vector Laboratories, Burlingame, CA, USA) after washing. Immunopositive cells were visualized by Vectastain Elite ABC Reagent (Vector Laboratories, Burlingame, CA, USA) with 0.05% 3,3-diaminobenzidine-4HCl (DAB, Nickel Solution) and 0.01% H_2_O_2_ as the chromogen.

### 2.5. Macrophage Cell Culture

RAW 264.7 macrophage cells were obtained from the American Type Culture Collection (ATCC; New York, NY, USA) and cultured in a humidified 5% CO_2_ incubator (Panasonic, Tokyo, Japan) at 37 °C using Dulbecco’s Modified Eagle Medium (DMEM, Hyclone, USA) supplemented with 10% fetal bovine serum (FBS), penicillin (100 U/mL), and streptomycin (100 μg/mL). For experimental purposes, the cells were routinely passaged to the third generation. The inflammation model was developed using RAW264.7 cells induced by lipopolysaccharide (LPS) in vitro. Water-soluble components are prepared by dissolving TJ107 to be tested in a serum-free RPMI1640 medium at 10 μg/mL, mixing at 37 °C for 30 min, and then centrifuging (3000 rpm/min, 15 min). The supernatant is sterilized by filtration through 0.22 m filters and diluted with basal medium to a final concentration of 1000 μg/mL for use. Cells were distributed into 4 groups: control group, TJ107 (100 µg/mL) group, LPS (400 ng/mL) group, and TJ107 + LPS group. The control group was maintained in a basal medium, the TJ107 group was cultured with TJ107, and the LPS group was stimulated with LPS to establish the inflammatory model. The TJ107 + LPS group were treated with TJ107 for 2 h and were stimulated with LPS for 4 h. After different treatments for 6 h, cells were harvested for real-time RT-PCR.

### 2.6. Analysis of Cytokine mRNA Using Real-Time RT-PCR

Total RNA was isolated from brain tissues (*n* = 5) or Raw264.7 cells (*n* = 4) using the Trizol reagent kit (Invitrogen Corp., Carlabad, CA, USA) according to the manufacturer’s protocol, and its concentration was calculated from the extinction at 260 nm, as determined spectrophotometrically. For cDNA synthesis, 10 μg of total RNA was reverse transcribed following the standard protocol using the High-Capacity cDNA Archive Kit (PE Applied Biosystems, Foster City, CA, USA). After that, 3 ng of cDNA was amplified by polymerase chain reaction (PCR) for 40 cycles and measured *IL-1β*, *IL-17*, *TNF-α*, *TLR2*, *TLR4*, *F4/80*, *CD-163*, *GFAP*, and *GAPDH*. Quantitative real-time PCR was performed in duplicate using the Thermal Cycler Dice Real-time System TP800 (TaKaRa, Tokyo, Japan). The Thermal Cycler Dice Real-time System software(version 4.02; Code TP900/TP960) (TaKaRa) was used to analyze the data, and the comparative *C*_t_ method (2∆∆*C*t) was used to quantify gene expression levels. The results are expressed relative to GAPDH. The relative mRNA intensities were calculated, and the expression of the control group was normalized to 1 at each point. The data was presented as mean ± standard deviation. Table 1 lists all the primers used in this analysis.

### 2.7. Statistical Analysis

Two-way ANOVA and Tukey’s multiple comparisons tests were used to analyze the differences between the multiple groups. A *p*-value < 0.05 was considered statistically significant.

## 3. Results

### 3.1. TJ107 Recovered Body Weight in MCAO Mice

Figure 1 displays the body weights of the mice (*n* = 10) in each experimental group from the seventh day after cerebral infarction surgery (Day 0) to Day 63. The mean body weight of the mice in the MCAO groups was significantly lower than that of the control mice at the start of the experiment (Day 0), probably due to surgery. After that the mean body weight of the MCAO mice was significantly lower than that of the control mice at the end of the experiment (Day 63), regardless of TJ107 supplementation (MCAO + TJ107 mice). The MCAO + TJ107 mice showed a significant increase in body weight from Day 14 compared to the MCAO mice and showed almost complete recovery to the same level as the control mice at Day 63. No differences were observed between the mice in the control and TJ107 groups.

### 3.2. TJ107 Significantly Reduced Infarct Volume at Day 60 After MCAO

To examine the therapeutic effect of TJ107 on the infarct volume after MCAO, continuous coronal slices sections from the center of the forebrain were subjected to H&E staining. For each section, the infarct volume was measured and analyzed using Image-j software. The observer was blinded when analyzing the slides. The infarction volume (μm^3^) was calculated as the average of the five sections and measured for each section. Compared to the MCAO group, the TJ107 treatment group showed a significant reduction in the infarct volume (Figure 2).

### 3.3. TJ107 Suppressed Brain-Infiltrating Macrophages and Reduced Expression of Immune Mediators in MCAO Mice

MCAO has been shown to induce an inflammatory process centered on macrophages during the acute phase and activate brain-infiltrating macrophages in a TLR 2/4-dependent manner, producing inflammatory cytokines such as IL-23. IL-23 induces IL-17 production by T cells [8]. Therefore, we examined the effects of TJ107 on macrophage infiltration and inflammatory response after MCAO and measured the levels of macrophage-specific mRNA species and some immune mediators in each group at Day 63 (Figure 2). It was found that the mRNA levels of *F4/80* (a marker of total macrophages), *CD163* (a marker of resident macrophages), *TRL2*, and *TRL4* were dramatically increased in the MCAO group; however, these levels were decreased in the MCAO + TJ107 group (Figure 3A). MCAO led to significant increases in *TNF-α*, *IL-1β*, and *IL-17* expressions compared to that in the controls, while TJ107 supplementation significantly reduced the MCAO-induced increases in all three genes, returning them to control levels. No differences were observed in the right side of the brain of the mice in each group.

Furthermore, in accordance with immunohistochemistry targeting F4/80, we investigated the macrophages infiltrating the left side of the brain in mice from each group. Compared to the results of the control group, the increased macrophage infiltration in the cerebral infarction site induced by MCAO was significantly inhibited by TJ107 (Figure 3B).

Furthermore, the mRNA levels of *IL-1β*, *IL-17*, *TNF-α*, *TLR2*, and *TLR4* in Raw264.7 cells cultured with or without TJ107 after LPS treatment were measured. LPS led to significant increases in *TNF-α*, *IL-1β*, *IL-17*, and *TRL2* expression compared to the no-treatment control, while TJ107 supplementation significantly reduced the LPS-induced increases in all four genes (Figure 3C).

### 3.4. TJ107 Suppresses GFAP Expression in MCAO Mice

To detect whether the TJ107 treatment in the MCAO mice decreased the astrocyte proliferation that leads to the extent of gliosis (glial scarring) around the lesion, staining of the infarcted area with the astrocyte marker GFAP showed that a glial scar formed between the normal and damaged areas, and astrocytes were stained a deep brown color (Figure 4A). At the same time, it was observed that the TJ107 treatment significantly (*p* < 0.05) suppressed the reaction of GFAP-expressing cells in the brain spheres compared with their counterparts in the MCAO group (Figure 4B), and the real-time RT-PCR data indicated that TJ107 administration significantly suppressed GFAP expression (*p* < 0.05) compared with that of the MCAO group (Figure 4C). This demonstrated that MCAO induced neuroinflammation, as evidenced by the increased GFAP staining intensity, and TJ107 could decrease brain inflammation by suppressing astrocyte reactivity caused by damage.

## 4. Discussion

Oriental medicine has been increasingly recognized as demonstrating therapeutic effects in the treatment of ischemic stroke [13]. In this study, we demonstrated that TJ107 had protective effects on chronic cerebral ischemic injury and may be a promising therapy for chronic ischemic stroke. These results suggest that, in the brain tissue of the MCAO region in the chronic phases, upregulated *TLR2* and *TLR4* expression and macrophage infiltration into the brain led to the production of inflammatory cytokines and promoted inflammation, possibly contributing to the expansion of the ischemic stroke infarct area and the worsening of neurological symptoms. On the other hand, the administration of TJ107 was proven to reduce the infarct area, with decreased GFAP expression, suppressed macrophage infiltration in the brain, and reduced inflammatory cytokine production compared with the MCAO mice. Taken together, it was indicated that TJ107 exerts an anti-inflammatory effect and is thought to promote functional recovery through reducing the extent of gliosis in the peri-infarct area.

When the brain is damaged by ischemia or other causes, neurons are the first to be affected. Their injury releases a plethora of DAMPs and neurotransmitters that can activate microglia and astrocytes [8,19]. This activation indicates the onset of neuroinflammation and is characterized by the release of proinflammatory cytokines, chemokines, and other inflammatory mediators [20,21]. Although post-ischemic inflammation is an unavoidable step in the progression of brain ischemia–reperfusion injury, excessive inflammatory responses with a strong damaging effect on nerve cells that initially survive and secondary damage to nerve tissue due to the spread of inflammation have been a concern. Clark and others discovered that doxycycline (a drug that suppresses white blood cell adhesion) reduces the size of infarction without decreasing the number of macrophages [22]. Additionally, activated microglia could damage the BBB and encourage immune cell infiltration, thereby exacerbating inflammation [23]. Our results demonstrated that, in the brain tissue of the MCAO mice in the chronic phases (70 days after the onset of cerebral infarction), upregulated *TLR2* and *TLR4* expression and increased macrophage infiltration into the brain led to the production of inflammatory cytokines and promoted inflammation. Therefore, neuroinflammation is considered an important target in the development of new stroke therapies, and the efficacy of anti-inflammatory treatments has been clarified in several studies. For example, microglia-derived TNF-α has been reported to mediate endothelial necroptosis and exacerbate damage to the blood–brain barrier (BBB) following ischemic stroke. In contrast, anti-TNF-α drugs significantly reduced BBB destruction and improved outcomes after the stroke [24]. Furthermore, post-ischemic treatment with the IL-1β antibody canakinumab significantly reduced infarct size and brain edema in MCAO mice, improving neurological performance [25]. In this study, we first demonstrated the therapeutic effect of administering the oriental medicine TJ107 to the MCAO mice in chronic ischemic stroke.

These proinflammatory molecules could then contribute to astrocyte activation [23]. Astrocytes are important for mitigating brain injury, reconstructing the BBB, and maintaining central nervous system homeostasis during the acute phase of ischemia [26,27]. However, in the chronic phase, astrocytes become activated and proliferate, causing gliosis (gliotic scarring) around the lesions, which hinders axonal regeneration in the area surrounding the infarction [28]. It is believed that reactive astrogliosis involves astrocyte hypertrophy, proliferation, and the upregulation of the intermediate filament GFAP [29]. This reaction protects nerve cells from harmful substances by isolating the damaged area while simultaneously inhibiting axon sprouting, which obstructs nerve regeneration [28,30,31,32]. It has been demonstrated that the removal of glial scars does not promote functional recovery, and that chondroitin sulfate proteoglycans 4 (*cspg4*) and *cspg5* are crucial for axonal growth following spinal cord injury [33]. Another important event is the activation of microglia, which jeopardizes the survival of neurons through the release of many inflammatory promoting factors and neurotoxic mediators [5,34,35]. However, currently, there are no effective treatments for post-stroke glial scar formation, which inhibits axonal outgrowth and functional recovery after stroke. Recent research has shown that astrocytic extracellular vesicles (AEVs) derived from reactive astrocytes with anti-inflammatory properties contain CD63 and CD9 tetraspanins and miR-146a-5p, which regulate glial scars by suppressing NF-κB and TNF-α, allowing axonal growth and improving recovery from stroke [36]. This study suggests that AEV delivering miR-146a-5p may have potential applications in stroke recovery; however, the treatment with AEV was conducted on the seventh day post-MCAO, which is the acute phase of ischemic stroke, and it improved stroke recovery by modulating glial scars after stroke and allowing axonal growth without reducing the size of the infarct [36]. In this study, we demonstrated that the TJ107 treatment in the MCAO mice decreased astrocyte proliferation that leads to the extent of gliosis (glial scarring) around the lesion, with decreased GFAP expression at 70 days after the onset of cerebral infarction, the chronic phase of ischemic stroke, accompanied with anti-inflammatory effects through TNF-α suppression. This is the first report describing TJ107 exerting a brain-protecting effect by suppressing abnormal astrocyte activation and is expected to be useful as a chronic cerebral infarction treatment.

There are various traditional medicines that could be used for treatment in the acute, subacute, or chronic phases of stroke. Such medicines include Woohwangchungsim Won (in Korean; Niuhuang Qingxin Pill in Chinese); Chunghyul-Dan (Qingxue Pill in Chinese); Angoongwoohwang Won (Angong Niuhuang Pill in Chinese); Gudeong San (Gouteng Powder in Chinese and Choto-San in Japanese); and Boyanghwano Tang (Buyang Huanwu Decoction in Chinese) [37,38,39,40,41,42]. However, there are several challenges when conducting experimental and clinical stroke studies using herbal materials, such as limited to no scientific evidence justifying their use to treat patients. There are studies indicating that several traditional medicines and their components show protective effects against cerebral ischemia, including brain edema and damage to the blood–brain barrier, in various animal models of cerebral ischemia [43,44,45,46,47,48,49,50]. These oriental medicines were administered from 2 h to 7 days after MCAO in rats or mice, the acute phase of ischemic stroke. In this experimental research, we selected animal models displaying similarities to clinical chronic ischemic stroke and detected the therapeutic effects of TJ107, which may have been due to its anti-inflammatory properties; thus, its development and use in clinical practice is recommended.

It is considered that the mean body weight of the MCAO mice was significantly lower than that of the control mice at the start of the experiment (Day 0) due to the surgery. Furthermore, this study demonstrated that the TJ107 treatment showed a significant increase in body weight compared to the MCAO mice and showed almost a complete recovery to the same level as the control mice at Day 63; it is probably thought that the MCAO condition is improving, and the mice regained their weight. In the next experiment, we will examine the pretext of significant weight recovery in mice to decrease the risk in patients who have suffered from a stroke if they supplement with TJ107. Furthermore, although there are no reports of toxicity of TJ107 to the cells, we will investigate the impact of changing the dose of TJ107 on cell viability using the WST-8 assay in the next experiment.

## 5. Conclusions

Axonal regeneration and tissue repair are important for functional recovery after cerebral infarction. It is hypothesized that neuroinflammation can be implicated in glial scar formation, which integrates to hinder or enhance axonal outgrowth in the chronic stage of stroke. This study demonstrated that the administration of Gosha-jinki-gan was proven to reduce the infarct area, with decreased GFAP expression, suppressed macrophage infiltration in the brain, and reduced *TNF-α*, *IL-1β*, and *IL-17* production compared with the MCAO mice. We aimed to develop a treatment that reduces neuroinflammation by modulating glial scars after a stroke. Anti-inflammation-mediated glial scar control is key to promoting functional recovery, and Gosha-jinki-gan therapy shows promise for this purpose.

## Figures and Tables

**Figure 1 biomolecules-15-01136-f001:**
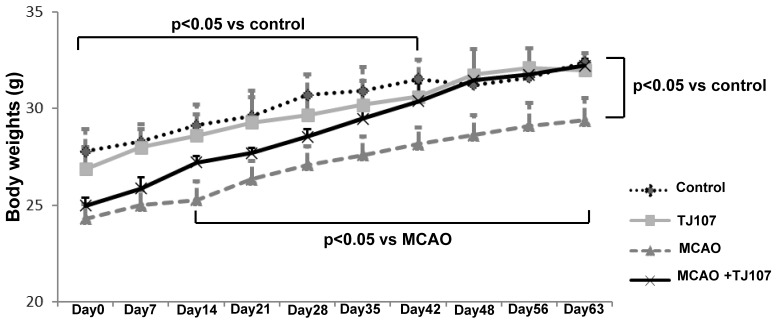
TJ107 recovered body weight in MCAO mice from Day 14 to Day 63. The data are presented as the mean ± standard deviation.

**Figure 2 biomolecules-15-01136-f002:**
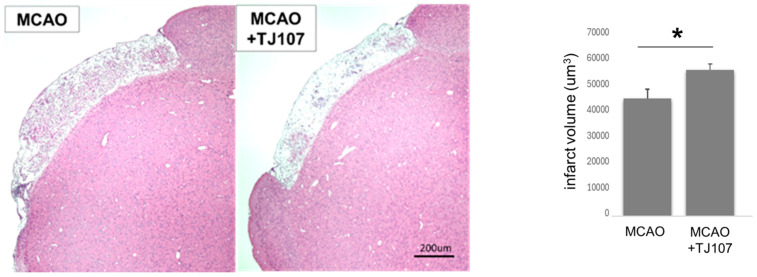
Infarct representative HE-stained coronal sections showing infarcts at Day 60 after MCAO in mice from the MCAO and MCAO + TJ107 groups. For each section, the infarct volume was measured and analyzed using Image-j software. The infarction volume (μm^3^, the extent of gliosis) was calculated as the average of the 5 sections. TJ107 supplementation significantly reduced infarct size at Day 60 after MCAO compared to untreated MCAO groups. Value is expressed as mean ± SD. * *p* < 0.05.

**Figure 3 biomolecules-15-01136-f003:**
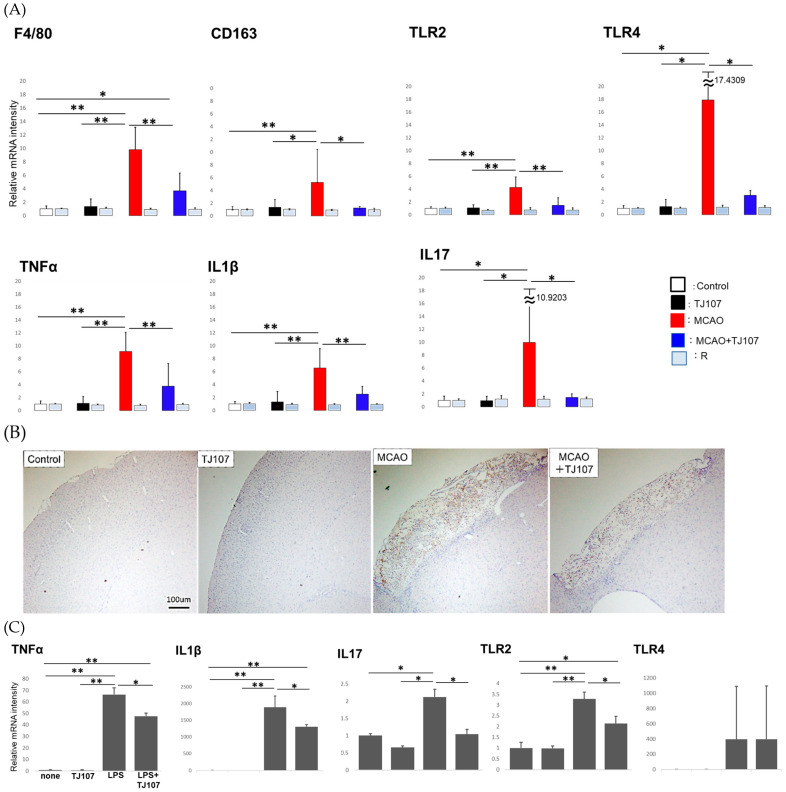
The effects of TJ107 on macrophage infiltration and inflammatory responses regulated by MCAO. (**A**) The mRNA levels of *IL-1β*, *IL-17*, *TNF-α*, *TLR2*, *TLR4*, *F4/80*, and *CD163* (macrophage marker) in brain tissues of MCAO mice compared to normal mice (*n* = 5). Expression was measured using real-time RT-PCR, and the results are expressed relative to the internal control *GAPDH*. The results are expressed as the mean ± SD of five mice in each group, and the *y*-axis shows the relative mRNA intensity. * *p* < 0.05 and ** *p* < 0.01. (**B**) F4/80 antigen-positive cells in brain sections (*n* = 5) of mice from the control, TJ107, MCAO, and MCAO + TJ107 groups. Many macrophages were found infiltrating the cerebral infarction site in the MCAO group and were suppressed in the MCAO + TJ107 group. Bar = 100 μm. (**C**) The mRNA levels of *IL-1β*, *IL-17*, *TNF-α*, *TLR2*, and *TLR4* in Raw264.7 cells cultured with or without TJ107 before LPS treatment (*n* = 4). Expression was measured using real-time RT-PCR, and the results are expressed relative to the internal control *GAPDH*. The results are expressed as the mean ± SD of five mice in each group, and the *y*-axis shows the relative mRNA intensity. * *p* < 0.05 and ** *p* < 0.01.

**Figure 4 biomolecules-15-01136-f004:**
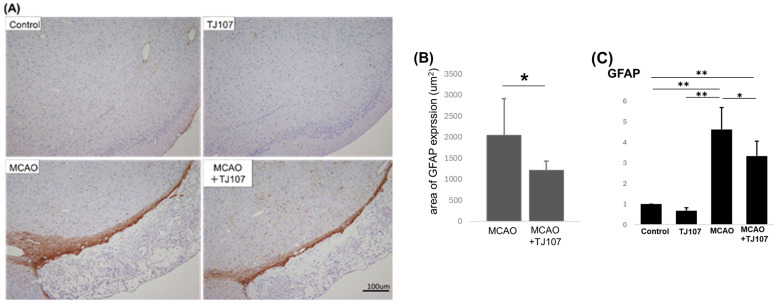
TJ107 suppresses GFAP expression in MCAO mice. (**A**) The amount of astrocytes displaying reactivity to glial fibrillary acidic protein (GFAP) increased in the MCAO mice compared with that in the MCAO + TJ107 group. (**B**) The area stained in deep brown was measured and analyzed using Image-j software. The stained area (μm^2^) was calculated as the average of the 4 sections. (**C**) The mRNA levels of *GFAP* in left-side brain tissues of MCAO mice compared to those from normal mice (*n* = 5). Expression was measured using real-time RT-PCR, and the results were expressed relative to the internal control *GAPDH*. The results are expressed as the mean ± SD of five mice in each group, and the *y*-axis shows the relative mRNA intensity. * *p* < 0.05 and ** *p* < 0.01.

**Table 1 biomolecules-15-01136-t001:** Primers used for real-time RT-PCR.

	Forward Primer (5′-3′)	Reverse Primer (5′-3′)
*GAPDH*	TGTGTCCGTCGTGGATCTGA	TTGCTGTTGAAGTCGCAGGAG
*TLR2*	TGCAAGTACGAACTGGACTTCT	CCAGGTAGGTCTTGGTCATT
*TLR4*	GTGCCATCATTATGAGTGCC	CAAGCCAAGAAATATACCATCGAAG
*IL-1β*	GAAGCTGGATGCTCTCATCTG	TTGACGGACCCCAAAAGAT
*IL-17*	CAACAGTAGCAAAGACTTGACCA	CCCAGGAAGACATACTTAGAAGAAA
*TNFα*	TCTTCTCATTCCTGCTTGTGG	TCTGGGCCATAGAACTGATGA
*F4/80*	CTTTGGCTATGGGCTTCCAGTC	GCAAGGAGGACAGAGTTTATCGTG
*CD163*	GGGTCATTCAGAGGCACAGTG	CTGGCTGTCCTGTCAAGGCT
*GFAP*	TGGCCACTGTGAGGCAGAAG	ACCTCCTCCTCGTGGATCTT

## Data Availability

The original contributions presented in this study are included in the article. Further inquiries can be directed to the corresponding author.

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
