# Peer review of "Gosha-Jinki-Gan Reduces Inflammation in Chronic Ischemic Stroke Mouse Models by Suppressing the Infiltration of Macrophages"

_biomolecules, 2025, doi:10.3390/biom15081136_

Round 1

Reviewer 1 Report

Comments and Suggestions for Authors

I am grateful for the opportunity to review the manuscript entitled "Gosha-jinki-gan reduces inflammation in chronic ischemic stroke mouse models by suppressing the infiltration of macrophages". I have carefully evaluated the manuscript, and here are my suggestions.

Introduction

The Introduction currently presents the temporal progression of ischemic stroke, acute, subacute, and chronic phases, in scattered and partially redundant paragraphs. This can make reading more tiring and less clear for the reader.

In the paragraph starting at line 31, the temporal progression is already well explained:

“The pathologic processes caused by stroke vascular injury occur in a time-dependent manner and are categorized as acute (minutes to hours), subacute (hours to days), and chronic (days to months)...”

However, in the following paragraphs, this explanation is repeated or restated in various forms (e.g., starting at line 40) without adding any fundamentally new information. This repeatedly reintroduces the same concept.

METHODOLOGY

Animals

The description of the MCAO model induction is superficial. Missing elements:

  • A detailed methodological reference for the surgical occlusion procedure (artery location, anesthesia used, temperature control, occlusion duration, etc.);
  • How was the occlusion confirmed (Doppler, clinical observation, or another method)?
  • Was mortality during or after surgery evaluated?

The MCAO model can show high variability, and its success rate strongly depends on the technique. A more precise description is essential.

Gosha-Jinki-Gan Diet

Clarify how the final dosage in mg/kg/day consumed by the animals was calculated.

Was the actual feed intake monitored? This may affect the dose received per animal.

Experimental Design

How was the sample size calculated?

There is no description of inclusion/exclusion criteria (e.g., animals with very small infarcts or those that did not survive until the end).

How many animals were effectively included in the final analysis? (For example, the n=5 mentioned for histological analysis must be justified.)

Indicate whether randomization and blinding were applied (ideally for data and histology analysis).

Histopathology

It is unclear at which brain level (bregma?) the stained sections were obtained. This is important for reproducibility.

Cell Culture

TJ107 was applied directly to the cultures, but how was the extract prepared for cell culture? (In what solvent? Filtered? Endotoxin-controlled?)

A cell viability control is missing (e.g., MTT or similar), to ensure cytokine inhibition was not due to TJ107 toxicity.

RT-qPCR

The final reaction volume and thermal cycle parameters are not described (only the equipment and kit are mentioned).

Statistics

The statistical software used is not specified.

It would be useful to know whether the data were tested for normality and homoscedasticity before conducting the ANOVA.

How were potential outliers handled?

RESULTS

There are no critical technical errors in the Results section, but several aspects require clarification or improvement to ensure robustness, reproducibility, and proper interpretation of the data.

TJ107 Recovered Body Weight in MCAO Mice:

Group n is not reiterated (presumably n=10, but should be specified again).

It is unclear whether the values represent group means ± SD or SEM, and whether repeated measures were applied (ideal for longitudinal data).

It would be helpful to indicate whether there were significant differences between MCAO and control mice from Day 0 or only post-surgery.

It is unknown whether animals with excessive weight loss were excluded (exclusion criteria?).

TJ107 significantly reduced infarct area at Day 60 after MCAO

Only 5 animals per group were used for histology. This should be justified (mortality? Randomization?).

Infarct was evaluated in 5 sections, but there is no description of the anatomical level (anterior-posterior bregma coordinates).

Calculating infarct volume (area × distance between sections) would be more robust.

Was the observer blinded when analyzing the slides?

TJ107 suppressed brain-infiltrating macrophages and reduced immune mediators

RT-PCR quantification was performed with n=5 for brain tissues and n=4 for cell culture, ideally, n≥6 to reduce biological variability.

In Figure 3C, TLR4 levels after LPS and TJ107 are not discussed — is there also a reduction?

  • For in vitro data:

The cell viability test is missing (to rule out TJ107 toxicity).

LPS + TJ107 treatment = TJ107 pre-treatment for 2 h + LPS for 4 h — it is important to confirm whether this time frame is sufficient for observing modulatory effects, not just suppression of basal gene expression.

TJ107 suppresses GFAP expression

GFAP is a good marker, but its expression varies regionally; the anatomical area of the analyzed sections should be specified.

Counting reactive astrocytes could be complemented with cell density or a semi-quantitative scoring system (instead of only stained area).

Other markers of astrogliosis or glial scar (e.g., vimentin or nestin) could be included, although GFAP alone is acceptable.

The manuscript states that TJ107 "reduced gliosis" — it would be more accurate to say that TJ107 "reduced GFAP expression and the extent of gliosis," as glial function was not tested.

DISCUSSION

The discussion is excessively long and repetitive

It spans about three full pages and includes:

Mechanisms already explained in the Introduction (e.g., the TLR2/4 → IL-23 → IL-17 inflammatory cascade);

A re-explanation of the results already described in the Results section.

Many references to previous studies with traditional medicine, not always directly related to the current findings.

Recommendation: The Discussion should be more concise, focused on the main findings, and contextualized with relevant literature, without redundancy. Consider reducing content by about 30%, grouping ideas, and avoiding repeated explanations.

1st paragraph of the Discussion:  This paragraph reintroduces detailed results already covered (e.g., increases in TLR2/4, IL-17), which may be repetitive.

It is too dense for the beginning of the Discussion section.

There is no dedicated paragraph for study limitations: Include a clear paragraph on limitations, preferably at the end of the Discussion, just before the Conclusions section.

Insert a proper transition to the Conclusions section.

Author Response

Responses to Reviewer 1’s comments:

I am grateful for the opportunity to review the manuscript entitled "Gosha-jinki-gan reduces inflammation in chronic ischemic stroke mouse models by suppressing the infiltration of macrophages". I have carefully evaluated the manuscript, and here are my suggestions.

Comments 1: Introduction

The Introduction currently presents the temporal progression of ischemic stroke, acute, subacute, and chronic phases, in scattered and partially redundant paragraphs. This can make reading more tiring and less clear for the reader.

In the paragraph starting at line 31, the temporal progression is already well explained:

“The pathologic processes caused by stroke vascular injury occur in a time-dependent manner and are categorized as acute (minutes to hours), subacute (hours to days), and chronic (days to months)...”

However, in the following paragraphs, this explanation is repeated or restated in various forms (e.g., starting at line 40) without adding any fundamentally new information. This repeatedly reintroduces the same concept.

Answer: Thank you for your precious advice. We have rewritten the part as “The main mechanisms of damage in cerebral infarction are different in the acute, subacute, and chronic phases, and currently, immune system activation is recognized as a major element in all stages of stroke pathophysiology, including long lasting regenerative processes [6,7]. The release of danger-associated molecular patterns (DAMPs), local expression of proinflammatory cytokines, subsequent expression of endothelial adhesion molecules, and breakdown of the blood–brain barrier (BBB) are among the initial events after arterial occlusion [7]. A study of a cerebral ischemia model revealed an inflammatory process centered on macrophages during the acute phase several days after the onset of cerebral infarction. In response to stimulation by DAMPs derived from dead cells, brain-infiltrating macrophages are activated in a toll-like receptor (TLR) 2/4-dependent manner, producing inflammatory cytokines such as IL-23. IL-23 induces IL-17 production from T cells that infiltrate later, promoting inflammation in the subacute phase, which contributes to the expansion of the infarct and the worsening of neurological symptoms [8].” (line 41-53). Thank you again for your advice.

Comments 2: METHODOLOGY

Animals

The description of the MCAO model induction is superficial. Missing elements:

  • A detailed methodological reference for the surgical occlusion procedure (artery location, anesthesia used, temperature control, occlusion duration, etc.);
  • How was the occlusion confirmed (Doppler, clinical observation, or another method)?
  • Was mortality during or after surgery evaluated?

The MCAO model can show high variability, and its success rate strongly depends on the technique. A more precise description is essential.

Answer: Thank you for your precious advice. MCAO mice used in this study were purchased from Clea Japan, Inc (Tokyo, Japan) and only succussed mice were used in experiment. According to the explanation from Clea Japan, Inc, we added the sentences about detailed methodological reference for the surgical occlusion procedure as “Briefly, the mice were anesthetized with halothane in a mixture of 70% nitrous oxide and 30% oxygen. The head temperatures maintained at 36 °C with a warming lamp and measured cerebral blood flow (CBF) before and during ischemia by laser Doppler flowmetry at the ipsilateral parietal bone (∼1−2 mm posterior to bregma). Each mouse's resting CBF value was regarded as the baseline for that mouse, and changes in CBF after the induction of brain ischemia were expressed as percentages of the resting value. The cerebral infarction model was created by directly applying electrical coagulation to the left MCA. During left MCA ligature, a reduction in CBF of more than 60% was confirmed by laser Doppler flowmetry”. (line 85-93). Thank you again for your advice.

Comments 3: Gosha-Jinki-Gan Diet

Clarify how the final dosage in mg/kg/day consumed by the animals was calculated.

Was the actual feed intake monitored? This may affect the dose received per animal.

Answer: Thank you for your precious advice. We have added the final dosage as “General condition, food intake (3g diet/mouse/day; 0.65mg TJ107/kg/day), and body weight were documented for all the mice at 7‐day intervals from 0 to 63 days after injection.” (line 117-118).

Comments 4: Experimental Design

How was the sample size calculated? There is no description of inclusion/exclusion criteria (e.g., animals with very small infarcts or those that did not survive until the end). How many animals were effectively included in the final analysis? (For example, the n=5 mentioned for histological analysis must be justified.)

Indicate whether randomization and blinding were applied (ideally for data and histology analysis).

Answer: Thank you for your precious advice. MCAO mice used in this study were purchased from Clea Japan, Inc (Tokyo, Japan) and no mice with very small infarcts or those that did not survive until the end. We have rewritten sentence as “MCAO mice arrived one week after day of surgery and were divided randomization into MCAO group (n = 10; mice that received surgery and were fed the standard MF diet until 63 days after arrival date, day 0) and MCAO +TJ107 group (n = 10; mice that received surgery and were fed the TJ107 diet until 63 days after arrival date, day 0).” (line 113-117). Thank you again for your advice.

Comments 5: Histopathology

It is unclear at which brain level (bregma?) the stained sections were obtained. This is important for reproducibility.

Answer: Thank you for your precious advice. We have rewritten sentence as “They were then embedded in paraffin and coronal slices sliced into 5 um sections” (line 126-127).

Comments 6: Cell Culture

TJ107 was applied directly to the cultures, but how was the extract prepared for cell culture? (In what solvent? Filtered? Endotoxin-controlled?) A cell viability control is missing (e.g., MTT or similar), to ensure cytokine inhibition was not due to TJ107 toxicity.

Answer: Thank you for your precious advice. We have added some sentence as “Water-soluble components are prepared by dissolving TJ107 to be tested in serum-free RPMI1640 medium at 10μg/ml, mixing at 37℃ for 30 minutes, and then centrifuging (3,000rpm/min, 15 minutes). The supernatant is sterilized by filtration through 0.22m filters and diluted with basal medium to a final concentration of 1,000μg/ml for use.” (line 151-155). Furthermore, although there are no reports of toxicity of TJ107 to the cells, we will investigate the impact of changing the dose of TJ107 on cell viability using the WST-8 assay in the next experiment, but we don't have enough time to do it at this time. Thank you again for your understanding.

Comments 7: RT-qPCR

The final reaction volume and thermal cycle parameters are not described (only the equipment and kit are mentioned).

Answer: Thank you for your precious advice. We have added some explanation sentence as “and its concentration was calculated from the extinction at 260 nm, as determined spectrophotometrically. For the first-strand cDNA synthesis,10 μg of total RNA was reverse-transcribed with a High Capacity cDNA Archive Kit (PE Applied Biosystems,Foster City, CA, USA) according to the standard protocol. Subsequently, 3 ng of cDNA was amplified by 40 cycles of polymerase chain reaction (PCR) to measure IL-1β, IL-17, TNF-α, TLR2, TLR4, F4/80, CD-163, GFAP, and GAPDH.” (line 165-170). Thank you again for your advice.

Comments 8: Statistics

The statistical software used is not specified. It would be useful to know whether the data were tested for normality and homoscedasticity before conducting the ANOVA.

How were potential outliers handled?

Answer: Thank you for your precious advice. We used “Two-way ANOVA and the Tukey's multiple comparisons test” in this study and the sentence is carelessmiss (line 182). Furthermore, the numbers of mice or cell used in this study are five or four, it seems inappropriate to test for normality and homoscedasticity.

Comments 9: RESULTS

There are no critical technical errors in the Results section, but several aspects require clarification or improvement to ensure robustness, reproducibility, and proper interpretation of the data.

TJ107 Recovered Body Weight in MCAO Mice: Group n is not reiterated (presumably n=10, but should be specified again). It is unclear whether the values represent group means ± SD or SEM, and whether repeated measures were applied (ideal for longitudinal data). It would be helpful to indicate whether there were significant differences between MCAO and control mice from Day 0 or only post-surgery. It is unknown whether animals with excessive weight loss were excluded (exclusion criteria?).

Answer: Thank you for your precious advice. We have added some sentence as “Figure 1 displayed the body weights of the mice (n=10) in each experimental group from the seventh day after cerebral infarction surgery (Day0) to Day63. The mean body weight of the mice in the MCAO groups was significantly lower than that of the control mice at the start of the experiment (Day0) probably due to surgery.” (line 186-189). No mice with excessive weight loss were excluded (exclusion criteria). Thank you again for your advice.

Comments 10: TJ107 significantly reduced infarct area at Day 60 after MCAO

Only 5 animals per group were used for histology. This should be justified (mortality? Randomization?). Infarct was evaluated in 5 sections, but there is no description of the anatomical level (anterior-posterior bregma coordinates). Calculating infarct volume (area × distance between sections) would be more robust. Was the observer blinded when analyzing the slides?

Answer: Thank you for your precious advice. We have rewritten sentence as “Continuous coronal slices sections from the center of the forebrain were subjected to hematoxylin and eosin (H&E) staining. For each section, the infarct volume was measured and analyzed using Image-j software. The observer blinded when analyzing the slides. The infarction volum (um3) was calculated as the average of the 5 sections and measured for each section. Compared to the MCAO groups, the TJ107 treatment group showed a significant reduction in the infract volume (Figure 2).” and corrected Figure 2. (line 199-204).

Comments11: TJ107 suppressed brain-infiltrating macrophages and reduced immune mediators

RT-PCR quantification was performed with n=5 for brain tissues and n=4 for cell culture, ideally, n≥6 to reduce biological variability. In Figure 3C, TLR4 levels after LPS and TJ107 are not discussed — is there also a reduction? For in vitro data: The cell viability test is missing (to rule out TJ107 toxicity).

LPS + TJ107 treatment = TJ107 pre-treatment for 2 h + LPS for 4 h — it is important to confirm whether this time frame is sufficient for observing modulatory effects, not just suppression of basal gene expression.

Answer: Thank you for your precious advice. We also think n≥6 is ideally but we don't have enough time to do it currently. So please accept the n=5 for brain tissues and n=4 for cell culture in this study.

As the answer as comments 6, I will investigate the impact of changing the dose of TJ107 on cell viability using the WST-8 assay in the next experiment. We conducted many preliminary experiments to determine the conditions for culturing the RAW 264.7 macrophage cells to confirm this time frame is sufficient for observing modulatory effects and then examined the RT-PCR quantification. Thank you again for your understanding.

Comments 12: TJ107 suppresses GFAP expression

GFAP is a good marker, but its expression varies regionally; the anatomical area of the analyzed sections should be specified. Counting reactive astrocytes could be complemented with cell density or a semi-quantitative scoring system (instead of only stained area).

Other markers of astrogliosis or glial scar (e.g., vimentin or nestin) could be included, although GFAP alone is acceptable. The manuscript states that TJ107 "reduced gliosis" — it would be more accurate to say that TJ107 "reduced GFAP expression and the extent of gliosis," as glial function was not tested.

Answer: Thank you for your precious advice. Staining of the infarcted area with the astrocyte marker GFAP showed that a glial scar formed between the normal and damaged areas and astrocytes were stained a deep brown colour, it is difficult to count reactive astrocytes so we only measured stained area. Furthermore, please accept the GFAP expression alone in this study. Thank you for your understanding. We have rewritten the sentences as “Taken together, it was indicated that TJ107 exerts an anti-inflammatory effect and is thought to promote functional recovery through reducing the extent of gliosis in the peri-infarct area.”(line 285-287) and “In this study, we demonstrated that the TJ107 treatment in the MCAO mice decreased astrocyte proliferation that leads to the extent of gliosis (glial scarring) around the lesion”(line 333-337). Thank you again for your advice.

Comments 13: DISCUSSION

The discussion is excessively long and repetitive

It spans about three full pages and includes: Mechanisms already explained in the Introduction (e.g., the TLR2/4 → IL-23 → IL-17 inflammatory cascade); A re-explanation of the results already described in the Results section. Many references to previous studies with traditional medicine, not always directly related to the current findings.

Recommendation: The Discussion should be more concise, focused on the main findings, and contextualized with relevant literature, without redundancy. Consider reducing content by about 30%, grouping ideas, and avoiding repeated explanations.

Answer: Thank you for your precious advice. We have rewritten the Discussion especially the sentences about mechanisms that already explained in the Introduction (e.g., the TLR2/4 → IL-23 → IL-17 inflammatory cascade) (line 275-355). Thank you again for your advice.

Comments 14: 1st paragraph of the Discussion:  This paragraph reintroduces detailed results already covered (e.g., increases in TLR2/4, IL-17), which may be repetitive. It is too dense for the beginning of the Discussion section.

Answer: Thank you for your precious advice. We have rewritten this paragraph as “Oriental medicine has been increasingly recog­nized as demonstrating therapeutic effects in the treatment of ischemic stroke [19]. In this study, we demonstrated that TJ107 had protective effects on chronic cerebral ischemic injury and may be a promising therapy for chronic ischemic stroke. These results suggest that, in the brain tissue of the MCAO region in the chronic phases, upregulated TLR2 and TLR4 expression and macrophage infiltration into the brain led to the production of inflammatory cytokines and promoted inflammation, possibly contributing to the expansion of the ischemic stroke infarct area and the worsening of neurological symptoms. On the other hand, the administration of TJ107 was proven to reduce the infarct area, with decreased GFAP expression, suppressed macrophage infiltration in the brain, and reduced inflammatory cytokines production compared with the MCAO mice. Taken together, it was indicated that TJ107 exerts an anti-inflammatory effect and is thought to promote functional recovery through reducing the extent of gliosis in the peri-infarct area.” (line 272-284).

Comments 15: There is no dedicated paragraph for study limitations: Include a clear paragraph on limitations, preferably at the end of the Discussion, just before the Conclusions section. Insert a proper transition to the Conclusions section.

Answer: Thank you for your precious advice. We have added a paragraph for study limitation as “It is considered that the mean body weight of the mice in the MCAO groups was significantly lower than that of the control mice at the start of the experiment (Day0) due to the surgery. Furthermore, this study demonstrated the TJ107 treatment showed a significant increase in body weight compared to the MCAO mice and showed almost complete recovery to the same level as the control mice at Day63, it is probably thought that the MCAO condition is improving, and mice regained their weight. In the next experiment we will examine the pretext of significant weight recover in mice to decrease the lisk in patients who have suffered from a stroke if they supplement with TJ107. Furthermore, although there are no reports of toxicity of TJ107 to the cells, we will investigate the impact of changing the dose of TJ107 on cell viability using the WST-8 assay in the next experiment.” (line 355-365).

Reviewer 2 Report

Comments and Suggestions for Authors

The manuscript by Mingli Xu et al. is interesting, well designed and presented. Some suggestions:

  1. What is the importance of the mouse weight in this experiment?
    2. Could you describe the results in Figures 2, 3, and 4 in more detail?
    3. I consider the experiments to be correct, but the description of the results is very limited. You should expand on the description of the results.
    4. In the discussion, could you link the results to information in the literature? Thank you.
  2.  

Author Response

Responses to Reviewer 2’s comments:

The manuscript by Mingli Xu et al. is interesting, well designed and presented. Some suggestions:

Comments 1: What is the importance of the mouse weight in this experiment?

Answer: Thanks for your precious advice. We have added some sentences as “It is considered that the mean body weight of the mice in the MCAO groups was significantly lower than that of the control mice at the start of the experiment (Day0) due to the surgery. Furthermore, this study demonstrated the TJ107 treatment showed a significant increase in body weight compared to the MCAO mice and showed almost complete recovery to the same level as the control mice at Day63, it is probably thought that the MCAO condition is improving, and mice regained their weight. In the next experiment we will examine the pretext of significant weight recover in mice to decrease the lisk in patients who have suffered from a stroke if they supplement with TJ107.” (line 355-361).

Comments 2: Could you describe the results in Figures 2, 3, and 4 in more detail?

Answer: Thanks for your precious advice and we have added some examination about Figure 2 as “Continuous coronal slices sections from the center of the forebrain were subjected to hematoxylin and eosin (H&E) staining. For each section, the infarct volume was measured and analyzed using Image-j software. The observer blinded when analyzing the slides. The infarction volum (um3) was calculated as the average of the 5 sections and measured for each section. Compared to the MCAO groups, the TJ107 treatment group showed a significant reduction in the infract volume (Figure 2).”. Furthermore, I have corrected Figure 3 and Figure 4 to be easier to read. Thank you again for your advice.

Comments 3: I consider the experiments to be correct, but the description of the results is very limited. You should expand on the description of the results.

Answer: Thanks for your precious advice. We have added some sentences as “The mean body weight of the mice in the MCAO groups was significantly lower than that of the control mice at the start of the experiment (Day0) probably due to surgery.”in the results of Figure 1. And added some examination about Figure 2 as “To examine the effect of TJ107 on the infarct volume after MCAO, continuous coronal slices sections from the center of the forebrain were subjected to hematoxylin and eosin (H&E) staining. For each section, the infarct volume was measured and analyzed using Image-j software. The observer blinded when analyzing the slides. The infarction volum (um3) was calculated as the average of the 5 sections and measured for each section. Compared to the MCAO groups, the TJ107 treatment group showed a significant reduction in the infarct volume (Figure 2).”. Furthermore, I have corrected Figure 3 and Figure 4 to be easier to read. Thank you again for your advice.

Comments 4: In the discussion, could you link the results to information in the literature?

Answer: Thanks for your precious advice. We have rewritten some sentences in Discussion as “Oriental medicine has been increasingly recog­nized as demonstrating therapeutic effects in the treatment of ischemic stroke [19]. In this study, we demonstrated that TJ107 had protective effects on chronic cerebral ischemic injury and may be a promising therapy for chronic ischemic stroke. These results suggest that, in the brain tissue of the MCAO region in the chronic phases, upregulated TLR2 and TLR4 expression and macrophage infiltration into the brain led to the production of inflammatory cytokines and promoted inflammation, possibly contributing to the expansion of the ischemic stroke infarct area and the worsening of neurological symptoms. On the other hand, the administration of TJ107 was proven to reduce the infarct area, with decreased GFAP expression, suppressed macrophage infiltration in the brain, and reduced inflammatory cytokines production compared with the MCAO mice. Taken together, it was indicated that TJ107 exerts an anti-inflammatory effect and is thought to promote functional recovery through reducing the extent of gliosis in the peri-infarct area.

When the brain is damaged by ischemia or other causes, neurons are the first to be affected. Their injury releases a plethora of DAMPs and neurotransmitters that can activate microglia and astrocytes [20, 21]. This activation marks the onset of neuroinflammation, characterized by the release of proinflammatory cytokines, chemokines, and other inflammatory mediators [22,23]. Although post-ischemic inflammation is an unavoidable step in the progression of brain ischemia–reperfusion injury, excessive inflammatory responses with a strong damaging effect on nerve cells that initially survive and secondary damage to nerve tissue due to the spread of inflammation have been a concern. Clark et al. found that, even though doxycycline (a drug that inhibits leukocyte adhesion) reduces the infarct size, it does not reduce the number of macrophages [24]. Additionally, activated microglia could damage the BBB and encourage immune cell infiltration, thereby exacerbating inflammation [25]. Our results demonstrated that, in the brain tissue of the MCAO mice in the chronic phases (70 days after the onset of cerebral infarction), upregulated TLR2 and TLR4 expression and increased macrophage infiltration into the brain led to the production of inflammatory cytokines and promoted inflammation. Therefore, neuroinflammation is considered an important target for the development of new stroke therapies, and the effects of anti-inflammatory treatment have been elucidated in several studies. For instance, it has been reported that microglia-derived TNF-αmediates endothelial necroptosis and exacerbates BBB disruption after ischemic stroke. In contrast, an anti-TNF-α drug significantly alleviated BBB destruction and improved stroke outcomes [26]. Moreover, post-ischemic treatment with the IL-1βantibody canakinumab markedly reduced the infarct size and cerebral edema and improved neurological performance in MCAO mice [27]. In this study, we first demonstrated that therapeutic effect about administering the oriental medicine TJ107 to the MCAO mice in chronic ischemic stroke.” (line 272-308). Thank you again for your advice.

Reviewer 3 Report

Comments and Suggestions for Authors

The authors present an interesting article exploring the impact of a traditional oriental supplement on inflammation markers using a MCAO mouse model as a proxy for chronic ischemic stroke.

Abstract. Consider rewording: “Current treatments for ischemic stroke are limited to effectiveness in the early stage.” To be more specific in the time frame and also the sentence isn’t as clear in its meaning.

Line 45: Please specify what you mean by “The release of danger molecules”. This is not a common colloquial term.

Line 57: replace “While” with “When”

Figure 1: Please state in the caption what the error bars represent (standard error, standard deviation) number of replicates (The number of mice tested is listed in the methods but adding it here as well would better inform the reader

Perhaps consider defining LPS as Lipopolysaccharide.

Figure 3A).  It is too small to read the numbers on the y axis.  The alignment of the key is off Based on the error bars I am surprised the P<0.01 for TNFalpha between the MCAO and MCAO+TJ107.  I am also confused that the P value is only P<0.05 and not P<0.01 for TLR4. Please double check confidence intervals.  Also what is the label for the Y axis?

Figure 3B) I cannot see the X axis labels for most of the graphs and it isnt’ aligned properly in A. It seems pixilated. Y-axis label?  This is a problem for RT-PCR data throughout.

Figure 4B&C) y-axis is grey and small and hard to read.  Also double check p values.  Surprised MCAO+TJ107 is not ** significant instead of just * significant as listed.

The discussion doesn’t go into the weight gain by the MCAO+TJ107 relative to MCAO mice but not in the control vs TJ107.  Could you explain why this might have occurred.  Is there risk that if the results are transferable to humans, is there a risk of significant weight gain in those who have suffered from a stroke if they supplement with TJ107.

The conclusions seem to make some broad jumps rather than just restating what was observed and suggest how to change it.

Author Response

Responses to Reviewer 3’s comments:

The authors present an interesting article exploring the impact of a traditional oriental supplement on inflammation markers using a MCAO mouse model as a proxy for chronic ischemic stroke.

Comments 1: Abstract. Consider rewording: “Current treatments for ischemic stroke are limited to effectiveness in the early stage.” To be more specific in the time frame and also the sentence isn’t as clear in its meaning.

Answer: Thanks for your precious advice and we have added some examination as “The pathologic processes caused by stroke vascular injury occur in a time-dependent manner and are categorized as acute, subacute and chronic. Current treatments for ischemic stroke are limited to effectiveness in the early stages.” (line 13-16). Thank you again for your advice.

Comments 2: Line 45: Please specify what you mean by “The release of danger molecules”. This is not a common colloquial term.

Answer: Thanks for your precious advice and we have rewritten it as “The release of danger-associated molecular patterns (DAMPs)” (new line 44) and “In response to stimulation by DAMPs derived from dead cells,” (new line 49). Thank you again for your advice.

Comments 3: Line 57: replace “While” with “When”

Answer: Thanks for your precious advice and we have corrected it (new line 55).

Comments 4: Figure 1: Please state in the caption what the error bars represent (standard error, standard deviation) number of replicates (The number of mice tested is listed in the methods but adding it here as well would better inform the reader

Answer: Thanks for your precious advice. We have corrected it as “Figure 1 displayed the body weights of the mice (n=10) in each experimental group from the seventh day after cerebral infarction surgery (Day0) to Day63.”(new line 186-187) and added a sentence as “The data are presented as the mean ± standard deviation.” (new line 197) in Figure 1. Thank you again for your advice.

Comments 5: Perhaps consider defining LPS as Lipopolysaccharide.

Answer: Thanks for your precious advice and we have corrected it (new line 150).

Comments 6: Figure 3A).  It is too small to read the numbers on the y axis. The alignment of the key is off Based on the error bars I am surprised the P<0.01 for TNFalpha between the MCAO and MCAO+TJ107.  I am also confused that the P value is only P<0.05 and not P<0.01 for TLR4. Please double check confidence intervals. Also what is the label for the Y axis?

Figure 3B) I cannot see the X axis labels for most of the graphs and it isnt’ aligned properly in A. It seems pixilated. Y-axis label? This is a problem for RT-PCR data throughout.

Answer: Thanks for your precious advice. We have added the y-axis label and corrected Figure 3 to be easier to read. We used Two-way ANOVA and the Tukey's multiple comparisons test to compared the TNFalpha between the MCAO and MCAO+TJ107 and p=0.0052. On the other hands, the SD about TLR4 is 17.4309 so p=0.0318. Thank you again for your advice.

Comments 7: Figure 4B&C) y-axis is grey and small and hard to read.  Also double check p values.  Surprised MCAO+TJ107 is not ** significant instead of just * significant as listed.

Answer: Thanks for your precious advice. We have corrected Figure 4 to be easier to read. And MCAO+TJ107 is ** significant and I have corrected it in Figure 4. It is a carelessmiss. Thank you again for your advice.

Comments 8: The discussion doesn’t go into the weight gain by the MCAO+TJ107 relative to MCAO mice but not in the control vs TJ107.  Could you explain why this might have occurred.  Is there risk that if the results are transferable to humans, is there a risk of significant weight gain in those who have suffered from a stroke if they supplement with TJ107.

Answer: Thanks for your precious advice. We have added a paragraph at Discussion as “It is considered that the mean body weight of the mice in the MCAO groups was significantly lower than that of the control mice at the start of the experiment (Day0) due to the surgery. Furthermore, this study demonstrated the TJ107 treatment showed a significant increase in body weight compared to the MCAO mice and showed almost complete recovery to the same level as the control mice at Day63, it is probably thought that the MCAO condition is improving, and mice regained their weight. In the next experiment we will examine the pretext of significant weight recover in mice to decrease the lisk in patients who have suffered from a stroke if they supplement with TJ107.”(line 355-362). Thank you again for your advice.

Comments 9: The conclusions seem to make some broad jumps rather than just restating what was observed and suggest how to change it.

Answer: Thanks for your precious advice. We have added sentences in Conclusion as “This study demonstrated that the administration of Gosha-jinki-gan was proven to reduce the infarct area, with decreased GFAP expression, suppressed macrophage infiltration in the brain, and reduced TNF-α, IL-1β, and IL-17 production compared with the MCAO mice. We aimed to develop a therapy to reduce neuroinflammation by regulating post-stroke glial scars. Anti-inflammation-mediated glial scar control is key to promoting functional recovery, and Gosha-jinki-gan therapy shows promise for this purpose.”(line 367-372). Thank you again for your advice.

Round 2

Reviewer 2 Report

Comments and Suggestions for Authors

Not

Author Response

Thank you again for your comments.